# Mental Health of Medical and Non-Medical Professionals during the Peak of the COVID-19 Pandemic: A Cross-Sectional Nationwide Study

**DOI:** 10.3390/jcm9082527

**Published:** 2020-08-05

**Authors:** Julian Maciaszek, Marta Ciulkowicz, Blazej Misiak, Dorota Szczesniak, Dorota Luc, Tomasz Wieczorek, Karolina Fila-Witecka, Pawel Gawlowski, Joanna Rymaszewska

**Affiliations:** 1Department of Psychiatry, Wroclaw Medical University, Pasteura 10 Str., 50-367 Wroclaw, Poland; julian.maciaszek@umed.wroc.pl (J.M.); marta.ciulkowicz@student.umed.wroc.pl (M.C.); tomasz.wieczorek@student.umed.wroc.pl (T.W.); karolina.fila-witecka@student.umed.wroc.pl (K.F.-W.); joanna.rymaszewska@umed.wroc.pl (J.R.); 2Department of Genetics, Wroclaw Medical University, Marcinkowskiego 1 Str., 50-368 Wroclaw, Poland; blazej.misiak@umed.wroc.pl; 3Practice of Family Doctors M.V. Domanscy, E. Gepperta 13 Str., 50-072 Wroclaw, Poland; dorota.luc@gmail.com; 4Department of Emergency Medical Service, Wroclaw Medical University, Parkowa 34 Str., 51-616 Wroclaw, Poland; pawel.gawlowski@umed.wroc.pl

**Keywords:** SARS-CoV-2, psychiatry, infectious disease, healthcare personnel, psychopathological symptoms

## Abstract

Background: The study aimed to compare psychopathological expressions during the COVID-19 (novel coronavirus disease 2019) pandemic, as declared on March 11th 2020 by the World Health Organization, with respect to which institutional variables might distinguish the impact of COVID-19 in medical and non-medical professionals. Methods: A cross-sectional study was performed nationwide between 16th March and the 26th April 2020 in Poland. A total of 2039 respondents representing all healthcare providers (59.8%) as well as other professionals filled in the sociodemographic section, the General Health Questionnaire-28 and the author’s questionnaire with questions related to exposure to severe acute respiratory syndrome coronavirus 2 (SARS-CoV-2) infection, the availability of protective measures, quarantine, change of working hours and place of employment during the pandemic, as well as feelings associated with the state of the pandemic. Results: Medical professionals more often presented with relevant psychopathological symptoms (GHQ-28 (General Health Questionnaire-28) total score >24) than the non-medical group (60.8% vs. 48.0%, respectively) such as anxiety, insomnia and somatic symptoms even after adjustment for potential confounding factors. Male sex, older age and appropriate protective equipment were associated with significantly lower GHQ-28 total scores in medical professionals, whereas among non-medical professionals, male sex was associated with significantly lower GHQ-28 total scores. Conclusions: Somatic and anxiety symptoms as well as insomnia are more prevalent among medical staff than workers in other professions. Targeting the determinants of these differences should be included in interventions aimed at restoring psychological well-being in this specific population. Apparently, there are present gender differences in psychological responses that are independent of profession.

## 1. Introduction

The novel coronavirus disease 2019 (COVID-19) was identified in Wuhan, China, in December 2019 and attributed to severe acute respiratory syndrome coronavirus 2 (SARS-CoV-2) infection. Subsequently, a rapid transmission of COVID-19 occurred across China and affected other countries. Although epidemics of infectious diseases have always had their place in history, this time, globalization has facilitated the spread of SARS-CoV-2, causing a pandemic, which was announced on March 11, 2020, by the World Health Organization (WHO). In addition, the WHO has estimated the COVID-19 mortality rate to be 3.4% [1]. Epidemiological studies have provided further evidence that the mortality rate increases with age and is associated with comorbid physical health impairments, especially those related to the cardiovascular system. Although the pandemic has largely changed research priorities, specific treatments and vaccines are not available yet [2]. Consequently, the COVID-19 outbreak has emerged as a global medical, social and economic threat. 

Apart from the direct consequences of COVID-19, it has been identified that the pandemic might have a great impact on mental health through various mechanisms. Firstly, it has been found that SARS-CoV-2 can impact the central nervous system, leading to acute psychiatric manifestations [3]. Secondly, social isolation and quarantine may trigger a number of maladaptive responses manifesting as post-traumatic stress symptoms, anxiety, fear, anger and confusion [4]. There is also evidence that quarantine conditions might have long-term effects on mental health [5]. It has been shown that individuals affected by the pandemic are struggling with the fear of uncertainty, death, loss of job, drastic changes of lifestyle, stigmatization, isolation, separation from family and beloved persons, disruption of the usual routine of life and grief [6]. The impact of the COVID-19 pandemic is also largely associated with the ongoing economic crisis, the loss of jobs and reduced revenues [7].

The severe psychological and physical impact on medical staff in terms of mental health outcomes has already been identified during previous epidemics [8,9,10]. Emerging evidence also indicates that medical staff might be particularly vulnerable to the negative effects of the COVID-19 pandemic [11]. Indeed, medical professionals standing on the front lines have direct contact with patients suspected of being infected. Consequently, many medical professionals became infected and some of them died [12]. In light of the growing mortality related to SARS-CoV-2 infection, long working time, a high level of uncertainty in the management of infected patients, healthcare workers are reporting increasing levels of anxiety associated with numerous clinical activities and present with symptoms of depression [13]. Although it has been observed that psychopathological expressions among medical professionals may differ from those observed in the general population, studies in this field have been performed with small samples and there is still a lack of nationwide studies [14]. In addition, several mechanisms underlying the specificity of psychopathological expressions among medical professionals need to be taken into consideration. These include various individual factors (e.g., age, sex and the presence of children) and institutional factors (e.g., the length of service, changes to working time and the availability of personal protective equipment).

Taking into account the limitations of previous studies and a number of research gaps, we aimed to compare psychopathological expressions during the COVID-19 pandemic in medical and non-medical professionals on the basis of a nationwide survey. Furthermore, we tested the hypothesis that there are various individual and institutional determinants of these responses that might distinguish the impact of COVID-19 on the psychological responses among two groups of professionals. 

## 2. Materials and Methods

### 2.1. Participants

Data were collected through an online survey administered between 16th March 2020 and the 26th April 2020 in Poland. The study was initiated 12 days after the first case of SARS-CoV-2 infection had been detected in Poland and covered the period of a rapid increase in the incidence of COVID-19 with subsequent social restrictions [15]. Participants over the age of 18 years were invited to participate in the survey that was distributed through social media and email addresses. The study was addressed to representatives of all medical professions as well as professions not related to healthcare. Participants representing the medical profession groups included doctors, nurses, paramedics, allied healthcare workers (pharmacists, physiotherapists, occupational therapists, and psychologists), technicians and administrators. Data analysis was limited to completed questionnaires. The study was approved by the Ethics Committee at the Wroclaw Medical University (Poland), and all participants provided written informed consent. The study was performed in accordance with the principles of the Declaration of Helsinki. 

### 2.2. Measures

The survey consisted of three sections: the sociodemographic section, the author’s questionnaire and the General Health Questionnaire-28 (GHQ-28). The sociodemographic questionnaire recorded data on general demographic characteristics such as age, sex, place of residence, marital status, education and profession. The author’s questionnaire was based on questions related to exposure to SARS-CoV-2 infection, the availability of protective measures, quarantine, change of working hours and the place of employment during the pandemic, as well as feelings associated with the state of the pandemic.

The GHQ-28 is a 28-item questionnaire used to identify minor psychiatric disorders in the general population, divided into four subscales. These are somatic symptoms (items 1, 3, 4, 8, 12, 14 and 16), anxiety and insomnia (items 2, 7, 9, 13, 15, 17 and 18), social dysfunction (items 5, 10, 11, 25, 26, 27 and 28) and severe depression (items 6, 19, 20, 21, 22, 23 and 24) [16]. The GHQ-28 items are based on the 4-point Likert scale (0—not at all, 1—no more than usual, 2—rather more than usual, and 3—much more than usual). The total score ranges between 0 and 84, where higher scores refer to higher levels of distress. The cut-off for clinical relevance was set at 24 points as described elsewhere [17]. 

### 2.3. Study Outcomes 

To evaluate the primary outcome variables, we measured the severity of psychopathological symptoms reported by the healthcare professionals and representatives of other professions during the survey administration period. Additionally, in order to obtain the secondary outcome variables, we investigated the association between individual and institutional factors and psychopathological outcomes assessed through the GHQ-28 score.

### 2.4. Statistical Analyses

The Mann-Whitney U test (for continuous variables) and chi-square or Fisher’s exact test (for qualitative variables) were used to compare groups. Due to multiple comparisons, the Bonferroni correction was applied to the level of significance. Taking into account 32 bivariate comparisons, the level of significance was finally set at 0.0016. Significant between-group differences in the levels of psychopathology after the Bonferroni correction were further tested using the analysis of co-variance (ANCOVA). The analysis of co-variance (ANCOVA) was performed to investigate the differences in the levels of psychopathological manifestations between medical and non-medical professionals after co-varying for potential confounding factors. Additionally, a linear regression model was prepared with the backward stepwise selection algorithm based on the Akaike information criteria. The model included continuous variables such as age and length of service and qualitative variables such as gender, protection against infection, major changes in private life, fear for personal health, fear for the health of loved ones, impact of media reports on mental state, frustration, loneliness because of isolation, anger, use of alcohol and nicotine and contact with COVID-19 without protective measures. The results of the ANCOVA and linear regression analysis were considered significant if the *p*-value was less than 0.05. All analyses were performed in R R Core Team (version 3.5.3, 2019, https://www.r-project.org/). 

## 3. Results

### 3.1. Participants

The general characteristics of the sample are presented in Table 1. Out of 2039 participants, 1216 (59.6%) individuals represented medical professions while 823 (40.4%) pursued non-medical occupations. The vast majority of respondents, regardless of career, were women (80.0% among medical professionals and 74.4% among non-medical professionals). Data were collected from respondents representing all administrative regions in Poland, and the majority of them (63.2%) represented very big cities (>300,000 inhabitants). The medical professionals included physicians (47.3%), nurses (16.5%), pharmacists (7.3%), laboratory diagnosticians (5.9%), dentists (5.3%), paramedics (4.9%), clinical psychologists or psychotherapists (3.5%), physiotherapists (3.3%), midwives (2%), secretaries or medical recorders (1.4%), occupational medical technicians (1.4%), dental assistants (0.7%), care assistants (0.4%), medical interns (0.1%) and occupational therapists (0.1%). The non-medical professionals included administrative staff and accountants (16.6%), teachers and lecturers (14.3%), service and trade workers (12.4%), Information Technology employees (11.7%), engineers and other highly qualified employees (9.7%), entrepreneurs (3.5%), people in managerial positions (3.4), manual workers (2.4%), scientists (1.9%), journalists (1.6%), social workers (1.3%), non-clinical psychologists or psychotherapists (1.3%), technical workers (1.2%), employees of uniformed services (0.5%) and others (18.2%). The number of females was significantly higher among the medical professionals. This group of participants was more likely to report an urban place of residence, caring for a disabled person, major changes in private life, working on a shift schedule, contact with a COVID-19 patient without personal protective equipment, contact with COVID-19 patients at work, work experience of death due to COVID-19 and too few employees compared to the workload. They were also less likely to report having children, work location change during the pandemic and appropriate protection against infection. Finally, medical professionals had significantly higher weekly working time and length of service.

### 3.2. Psychopathological Outcomes 

Medical professionals more often met the criterion for the presence of relevant psychopathological symptoms (a GHQ-28 total score > 24) than the non-medical group (60.8% vs. 48.0%, respectively). Moreover, they had also significantly higher GHQ-28 scores (all subscales and the total score) than the other participants (Table 2). The observed statistical power for detecting between-group differences in the GHQ-28 scores was as follows: 64.8% for severe depression, 100% for somatic symptoms, 100% for anxiety and insomnia, 64.5% for social dysfunction and 100% for the GHQ-28 total score. The ANCOVA revealed a significant effect of group (medical vs. non-medical professionals) on the level of somatic symptoms, anxiety and insomnia as well as the GHQ-28 total score after co-varying for the effects of potential confounding factors (Table 3). There were significant independent effects of sex in all the ANCOVA models. The effect of age appeared to be significant in the ANCOVA model testing that included the GHQ-28 total score, the anxiety and insomnia domain and the depression domain as a dependent variable. In turn, the effect of having children was independently negatively associated with the depression score, while the reports of caring for a disabled person were significantly associated with the GHQ-28 score for somatic symptoms. There was also a significant and independent effect of shift work on the score for the somatic symptoms domain. Finally, the effect of group appeared to be non-significant in the ANCOVA models that included the GHQ-28 scores for social dysfunction and depression.

### 3.3. Determinants of Psychopathological Outcomes in Medical and Non-Medical Professionals

The results of the linear regression analysis testing for the factors related to the GHQ-28 total scores in the medical and non-medical professionals are shown in Table 4. Male sex, older age and appropriate protection against infection were associated with significantly lower GHQ-28 total scores in medical professionals. In turn, fear for the health of loved ones was associated with significantly higher GHQ-28 total scores in this group of participants. Among both groups, major changes in private life, fear for personal health, following media reports, frustration, loneliness, anger and increased use of alcohol and nicotine were also significantly associated with higher GHQ-28 total scores. In non-medical professionals, contact with a COVID-19 patient without personal protection equipment was correlated with significantly higher GHQ-28 total scores. Male sex was associated with significantly lower GHQ-28 total scores in participants involved in non-medical professions.

## 4. Discussion

Our study indicates the occurrence of maladaptive psychological responses to the COVID-19 pandemic among medical workers in comparison to that in people performing other professions in Poland. The findings from this survey imply that healthcare professionals present with higher levels of psychopathological symptoms in terms of anxiety, insomnia and somatic symptoms than those representing other professions, even after adjustment for potential confounding factors. To our knowledge, this is the first study comparing medical and non-medical professionals in terms of psychopathological manifestation during the COVID-19 pandemic. Over 60% of medical professionals and 48% of individuals working in non-medical professions from the study sample presented clinically relevant psychopathological symptoms. These findings are similar to those reported by a recent population-based study in China that reported symptoms of depression, anxiety, distress and insomnia in 34.0–71.5% of medical workers [18]. Similarly, another study reported that 63% of medical workers in Wuhan, China, demonstrated various psychopathological symptoms [19]. However, a lower prevalence of psychopathological symptoms compared to in our study was observed by the authors of the recent cross-sectional survey study based on over 4000 healthcare workers from Wuhan in which 39.1% of the study participants had psychological distress [20]. Lai et al. suggested that nurses, women, frontline medical workers and those working in Wuhan, China, were more likely to report various psychopathological symptoms [18], which is consistent with our findings in the relation to female sex. The vast majority of our results confirm observations from Asian countries during the initial stages of the COVID-19 pandemic [2,21]. Recently, there have been only a few reports defining the role of factors affecting the development of psychiatric symptoms in the pandemic [19,22,23,24]. However, there is still a lack of research identifying institutional and individual risk and protective factors affecting the mental health of healthcare workers and other citizens during the pandemic.

This study emphasizes that one of the most important institutional factors that affects mental health is the provision to medical workers of a sense of security in the workplace. The results point to the importance of appropriate protection against infection as the main mental-health-related factor during the pandemic that affects all the domains. This is in accordance with recent studies related to medical staff, which identify access to personal protective equipment as an independent predictor of a lower level of mental distress [25,26] as well as one of the main concerns of healthcare workers [20]. It seems that these results are not revealing; however, at the same time, our findings show that the vast majority of staff deem the institution’s activities in providing security to be insufficient. This is likely not unique to Poland, as recent studies have also found a lack of personal protective equipment being reported by medical health workers across other countries [27,28,29]. Furthermore, the present study highlights that the sense of security could be considered from different perspectives. Both groups of medical and non-medical professionals revealed anxiety about the state of their health. This is consistent with the cross-sectional study performed in China in which the authors suggested the fear of being infected to be a risk factor for mental distress [30]. However, this study highlights another important factor, which is the fear for loved ones, that was visible only among medical professions. Medical workers remain with an internal dilemma related, on the one hand, to a sense of loyalty to the profession and their patients and, on the other hand, to the responsibility for their families [31]. This is confirmed by the recent study from Wuhan in which the authors demonstrated that the majority of healthcare workers were concerned about the infection of family members [20]. This kind of long-lasting internal emotional tension might be manifested in psychopathological symptoms among most medical workers during a pandemic, which has already been observed in 2003 during the outbreak of severe acute respiratory syndrome (SARS) [32] and in 2014–2015 during the Ebola outbreak [33]. Despite the discussed fear regarding the infection of family members among medical professionals on the one hand, we emphasized the protective effect of having children on the development of depressive symptoms and, on the other hand, the relationship between care for an elderly person and the severity of psychopathological symptoms. From an individual-level perspective, this study indicates that men were less prone to the presence of psychopathological symptoms. In our study, male sex appeared to be negatively associated with total GHQ-28 scores, which was observed among both medical and non-medical professionals. These reports are similar to the results of recent studies performed in China in which being female was considered a significant risk factor for the development of severe depressive and anxiety symptoms, and distress [18,22]. 

We emphasized that following media reports was a risk factor for developing psychopathological symptoms among both groups. Our results correspond with another Wuhan online survey study [34] where spending over 2 h checking COVID-19-related information via social media was correlated with anxiety and depressive symptoms. The issue of the impact of excessive searching for COVID-19 news on mental health is particularly up to date according to recent studies, which confirm that the pandemic affected the content searched on the internet [35,36].

We observed that medical professionals more often than other respondents suffered from somatic symptoms as well as anxiety and insomnia. A higher prevalence of somatic symptoms during stressful situations, such as work in outbreak conditions, can be considered a physiological reaction caused by increased activity of the autonomic nervous system. Although a short-term hyperactivity of the sympathetic nervous system does not lead to any serious health-related consequences, the prolonged hyperactivity of the stress-related hypothalamic-pituitary-adrenal axis might lead to fatigue, depression, and other health-related outcomes [37,38,39]. As demonstrated by studies on previous outbreaks [40,41], some of the medical workers during the SARS-CoV-2 pandemic may be at risk for post-traumatic stress disorder, which also appears to be connected with prolonged hypothalamic–pituitary–adrenal (HPA) axis overactivity [42,43]. From a psychodynamic perspective, prolonged emotional tension can lead medical workers to channel difficult emotional experiences into somatic symptoms and insomnia, which are easier for them to accept than developing depressive symptoms that may lead to an occupational dysfunction and could be understood as the effect of defense mechanisms.

There are some limitations of this study that need to be discussed. Firstly, we did not record the initial number of individuals approached for participation and the reasons for non-participation were not recorded. Therefore, the representativeness of the sample is limited. Another point is that the assessment of psychopathological symptoms was limited to the use of GHQ-28, and thus, we were not able to record specific diagnoses. It should also be noted that our survey was not administered longitudinally. In this regard, the temporal patterns of psychopathological expressions were not addressed. Another limitation is response bias due to the online form of the questionnaire distribution.

In summary, our study provides evidence that medical professionals are more vulnerable to developing anxiety, insomnia and somatic symptoms in response to the pandemic. In addition, the determinants of psychopathological expressions in these two groups differ in terms of age, care for an elderly or disabled person, contact with COVID-19 at work and contact with COVID-19 without protection measures. Apparently, there are present gender differences in psychological responses that are independent of the profession.

Nevertheless, these findings create grounds for personalizing interventions that aim to restore psychological wellbeing in medical and non-medical professionals as well as emphasizing key factors affecting the greater susceptibility for a negative psychological response during the pandemic, some of which are modifiable.

## Figures and Tables

**Table 1 jcm-09-02527-t001:** General characteristics of medical professionals and individuals representing non-medical professionals.

	Medical Professionals, *n* = 1216	Non-Medical Professionals, *n* = 823	*p*-Value
Sex, females	973 (80.0%)	612 (74.4%)	0.003
Age, years	39.23 (12.26)	40.4 (13.24)	0.093
Urban place of residence	1177 (96.79%)	756(91.86%)	<0.001
In relationship or married	934 (76.8%)	619 (75.2%)	0.437
Having children	622 (51.2%)	468 (56.9%)	0.013
Caring for a disabled person	219 (18.1%)	100 (12.2%)	<0.001
Major changes in private life	229 (24.1%)	115(18%)	0.011
Working time (hours per week)	44.89 (14.27)	39.15 (11.18)	<0.001
Length of service (years)	14.59 (12.53)	18.84 (11.94)	<0.001
Work location change during the COVID-19 pandemic	359 (29.9%)	418 (62.1%)	<0.001
Increase in working time	218 (19.7%)	157 (25.7%)	0.041
Work in a shift system	507 (42.4%)	56 (8.4%)	<0.001
Contact with the COVID-19 patient without personal protective equipment	207 (17%)	41 (5%)	<0.001
Confirmed COVID-19 infection	12 (1%)	3 (0.4%)	0.121
Contact with the COVID-19 patients at work	289 (24.1%)	9 (1.3%)	<0.001
Work experience of death due to COVID-19	41 (3.4%)	7 (1%)	0.003
Appropriate protection against infection	356 (29.7%)	522 (77.9%)	<0.001
Too few employees compared to the workload	798 (66.4%)	236 (35.5%)	<0.001

COVID-19, novel coronavirus disease 2019. Data expressed as *n* (%) or mean (SD).

**Table 2 jcm-09-02527-t002:** Measures of psychopathology in medical professionals and individuals representing non-medical professions.

	Medical Professionals, *n* =1216	Non-Medical Professions, *n* =823	*p*-Value
GHQ-28—total score	29.7 (14.9)	26.1 (14.8)	<0.001
GHQ-28—positive scoring	739 (60.8%)	395(48.0%)	< 0.001
GHQ-28—somatic symptoms	7.7 (4.6)	6.5 (4.5)	<0.001
GHQ-28—anxiety and insomnia	10.0 (5.4)	8.2 (5.3)	<0.001
GHQ-28—social dysfunction	8.5 (3.5)	8.2 (3.5)	0.037
GHQ-28—severe depression	3.5 (3.9)	3.2 (3.9)	0.036
The use of sedatives	177(14.8%)	85 (12.7%)	0.234
Fear for personal health	679 (55.9%)	328 (39.9%)	<0.001
Fear for the health of loved ones	714 (58.7%)	426 (51.8%)	<0.001
Worsening of mental health due to media reports	811 (66.7%)	482 (58.5%)	<0.001
Frustration	990 (81.4%)	612 (74.4%)	<0.001
Loneliness because of isolation	754 (62%)	501 (60.9%)	0.138
Anger	919 (75.6%)	521 (63.3%)	<0.001
Increased alcohol or nicotine intake	305 (25.1%)	145 (17.6%)	0.002

GHQ-28, General Health Questionnaire-28. Data expressed as *n* (%) or mean (SD).

**Table 3 jcm-09-02527-t003:** Psychopathological expressions in medical and non-medical professionals after adjustment for potential confounding factors.

	GHQ-28 Total Score		GHQ-28 Somatic Symptoms		GHQ-28 Anxiety and Insomnia	
	F	*p*	F	*p*	F	*p*
Medical/non-medical profession	13.877	<0.001	12.678	<0.001	25.988	<0.001
Sex	77.337	<0.001	88.272	<0.001	93.801	<0.001
Age	6.438	0.011	3.757	0.053	5.383	0.020
Place of residence	0.675	0.411	2.720	0.099	0.666	0.415
Children	0.983	0.322	0.006	0.939	0.040	0.841
Caring for a disabled person	3.396	0.066	4.448	0.035	2.515	0.113
Change in working time	1.406	0.236	1.003	0.317	0.613	0.434
Shift work	3.225	0.073	5.279	0.022	3.122	0.077

Significant effects (*p* < 0.05) are marked in bold.

**Table 4 jcm-09-02527-t004:** Factors related to the GHQ-28 total scores in medical and non-medical professionals (results of linear regression analysis).

Group of Participants	Variable	Beta	*p*-Value	VIF	95% CI
Medical professionals	male sex	−4.789	<0.000	1.11	−6.520–3.058
	age	−0.047	0.022	1.93	−0.121–0.028
	urban place of residence	2.266	0.239	1.03	−1.507–6.039
	in relationship or married	−0.196	0.816	1.12	−1.851–1.459
	having children	−0.812	0.375	1.84	−2.607–0.983
	caring for a disabled person	1.379	0.136	1.12	−0.435–3.193
	professional inactivity	0.428	0.861	1.08	−4.365–5.220
	contact with COVID-19 without protective measures	0.798	0.395	1.11	−1.042–2.639
	confirmed SARS-CoV-2 infection	3.995	0.256	1.08	−2.908–10.897
	confirmed SARS-CoV-2 infection among family or friends	−1.561	0.235	1.07	−4.142–1.019
	death due to COVID-19 among family or friends	−0.429	0.929	1.02	−9.857–8.998
	contact with people with COVID-19 at work	1.075	0.202	1.14	−0.578–2.727
	experience of death from COVID-19 in the workplace	0.716	0.712	1.10	−3.092–4.524
	appropriate protection against infection	−1.742	0.029	1.18	−3.309–0.174
	major changes in private life	2.916	<0.000	1.10	1.284–4.547
	fear for my health	6.290	<0.000	1.46	4.681–7.899
	fear for the health of loved ones	2.926	<0.000	1.41	1.332–4.521
	media reports worsen mental state	3.224	<0.000	1.48	1.522–4.927
	frustration	4.251	<0.000	1.46	2.202–6.299
	loneliness because of isolation	2.841	<0.000	1.24	1.323–4.359
	anger	2.708	0.003	1.32	0.927–4.489
	increased alcohol of nicotine intake	6.127	<0.000	1.08	4.549–7.706
Non-medical professions	male sex	−2.583	0.018	1.11	−4.718–0.448
	age	0.070	0.146	1.45	−0.025–0.164
	urban place of residence	−1.297	0.468	1.02	−4.804–2.210
	in relationship or married	−1.509	0.209	1.20	−3.866–0.849
	having children	−0.928	0.453	1.70	−3.359–1.502
	caring for a disabled person	−1.555	0.263	1.09	−4.281–1.171
	professional inactivity	−5.106	0.142	1.11	−11.935–1.722
	contact with COVID-19 without protective measures	4.637	0.029	1.10	0.472–8.802
	confirmed SARS-CoV-2 infection	12.555	0.150	1.09	−4.554–29.663
	confirmed SARS-CoV-2 infection among family or friends	1.149	0.574	1.10	−2.859–5.157
	death due to COVID-19 among family or friends	1.682	0.760	1.08	−9.128–12.491
	contact with people with COVID-19 at work	−0.755	0.852	1.04	−8.677–7.167
	experience of death from COVID-19 in the workplace	−8.593	0.063	1.06	−17.658–0.473
	appropriate protection against infection	−0.136	0.906	1.10	−2.404–2.131
	major changes in private life	4.014	0.003	1.18	1.381–6.646
	fear for my health	7.371	<0.000	1.58	5.075–9.666
	fear for the health of loved ones	1.250	0.245	1.38	-0.862–3.362
	media reports worsen mental state	3.827	0.001	1.40	1.667–5.988
	frustration	5.088	0.000	1.47	2.586–7.591
	loneliness because of isolation	3.673	0.001	1.33	1.552–5.794
	anger	3.454	0.001	1.27	1.350–5.557
	increased alcohol of nicotine intake	5.324	0.000	1.04	3.016–7.632

SARS-CoV-2, Severe acute respiratory syndrome coronavirus 2.

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
