# Peer review of "Mental Health of Medical and Non-Medical Professionals during the Peak of the COVID-19 Pandemic: A Cross-Sectional Nationwide Study"

_jcm, 2020, doi:10.3390/jcm9082527_

Round 1
Reviewer 1 Report
The authors specify a general hypothesis. The should be more specific in respect of which variables might distinguish the impact of Covid in medical versus non-medical workers. Line 11 minor "type Chi square "of" should be chi square "or" Was any power calculation carried out? In the list of professions/workers included domestic and care assistants are not mentioned. Was this because they were included under some other group? Because there is multiple testing, a Bonferroni correction should have been considered. If not, why not? Confidence intervals are preferable to p values Linear regression is used - was any test for normality of distribution or multicollinearity conducted since both are requirements for linear modelling. Line 268- "might" should be replaced by "is" in relation to the impact on generalisability. Considering other weaknesses another is likely to be response bias, since this was an online study.Author Response
Dear Reviewer,
Please see the attachment.

Reviewer 2 Report
The present manuscript is really of interest in the field of mental health and COVID-19 pandemia.
However, I consider that prior to the publication of these results, several aspects should be improved. Minor changes have been recommended, all of them detailed according to the sections of the manuscript.
Abstract section:
-I would also add in the conclusions subsection that apparently, gender differences in psychological responses have been found in the studied populations. The authors reported that male sex was associated with lower scores in the GHQ scale.
Introduction:
I would prefer to refer to psychological responses or psychopathological expression (of symptoms) rather than psychopathological responses. This may apply for the rest of the manuscript.
Methods:
2.3.) I would better define primary and secondary outcome variables in this subsection.
2.4.) Statistical analyses. I would prefer to refer to continuous variables rather than numerical when defining which variables were compared by means of the Mann-Whitney U test. This may be in contrast the qualitative variables which were analysed by the chi-square or Fisher exact test.
I would also add which variables were introduced into the linear regression models, in the statistical analysis section. Were these variables continuous?
Results:
3.1. Participants. I would also define what are the authors refering to "non-medical occupations". Which individuals are classified within this group?
Table 1. I would correct the terms "non-medicals professionals" instead of "non-medical professions"
Table 4. It should be better indicated which kind of statistical analyses were described in this table.
Discussion
I would better describe the appearance or not of gender differences in psychological symptoms.
Round 2
Reviewer 1 Report
The authors have addressed all my comments